# Benchmarking of Approaches for Gene Copy-Number Variation Analysis and Its Utility for Genetic Aberration Detection in High-Grade Serous Ovarian Carcinomas

**DOI:** 10.3390/cancers16193252

**Published:** 2024-09-24

**Authors:** Pavel Alekseevich Grebnev, Ivan Olegovich Meshkov, Pavel Viktorovich Ershov, Antonida Viktorovna Makhotenko, Valentina Bogdanovna Azarian, Marina Vyacheslavovna Erokhina, Anastasiya Aleksandrovna Galeta, Aleksandr Vladimirovich Zakubanskiy, Olga Sergeevna Shingalieva, Anna Vasilevna Tregubova, Aleksandra Vyacheslavovna Asaturova, Vladimir Sergeevich Yudin, Sergey Mihaylovich Yudin, Valentin Vladimirovich Makarov, Anton Arturovich Keskinov, Anna Sergeevna Makarova, Ekaterina Andreevna Snigir, Veronika Igorevna Skvortsova

**Affiliations:** 1Federal State Budgetary Institution “Centre for Strategic Planning and Management of Biomedical Health Risks” of the Federal Medical Biological Agency (Centre for Strategic Planning of FMBA of Russia), Bld. 1, Pogodinskaya Street, 10, 119121 Moscow, Russia; pgrebnev@cspfmba.ru (P.A.G.); imeshkov@cspfmba.ru (I.O.M.); pershov@cspfmba.ru (P.V.E.); amahotenko@cspfmba.ru (A.V.M.); vazaryan@cspfmba.ru (V.B.A.); merokhina@cspfmba.ru (M.V.E.); agaleta@cspfmba.ru (A.A.G.); azakubanskiy@cspfmba.ru (A.V.Z.); oshingalieva@cspfmba.ru (O.S.S.); vyudin@cspfmba.ru (V.S.Y.); yudin@cspfmba.ru (S.M.Y.); makarov@cspfmba.ru (V.V.M.); amakarova@cspfmba.ru (A.S.M.); 2Federal State Budgetary Institution “National Medical Research Center for Obstetrics, Gynecology and Perinatology Named after Academician V.I. Kulakov”, Ministry of Healthcare of The Russian Federation, Oparina Street, Bld. 4, 117997 Moscow, Russia; annyupitrue@mail.ru (A.V.T.); velikova@yandex.ru (A.V.A.); 3Federal Medical Biological Agency (FMBA of Russia), Volokolamskoye Shosse, 30, 123182 Moscow, Russia; skvortsova@cspfmba.ru

**Keywords:** CoreExome DNA microarray, NanoString CNV panel, digital droplet PCR (ddPCR), benchmarking, gene copy number variation, ovarian cancer, PABAK, high-grade serous carcinoma, HGSC

## Abstract

**Simple Summary:**

Our research integrates two key areas of focus. We first focus on a comparative biostatistical analysis of the agreement between three laboratory methods for genotyping gene copy number variations, and then we explore new characteristics of the distribution of this type of genetic aberration among cancer-associated genes. The results could be used to help molecular oncologists pick the smallest set of methods that will allow for accurate CNV genotyping in personalized cancer prognosis and drug-response prediction. We see the following strengths of our research: (1) We employ various methods for the detection of CNVs, such as CoreExome microarrays, nCounter v2 Cancer CN Assay panel, and digital droplet PCR; (2) This study sets out an interpretable comprehensive biostatistical analysis of the agreement using the PABAK score and Passing–Bablok regression; (3) This study presents the detection of CNVs in 12 paired tumor ovarian cancer/normal samples with clinical data; and (4) The research has translational potential.

**Abstract:**

**Objective**: The goal of this study was to compare the results of CNV detection by three different methods using 13 paired carcinoma samples, as well as to perform a statistical analysis of the agreement. **Methods**: CNV was studied using NanoString nCounter v2 Cancer CN Assay (Nanostring), Illumina Infinium CoreExome microarrays (CoreExome microarrays) and digital droplet PCR (ddPCR). **Results**: There was a good level of agreement (PABAK score > 0.6) between the CoreExome microarrays and the ddPCR results for finding CNVs. There was a moderate level of agreement (PABAK values ≈ 0.3–0.6) between the NanoString Assay results and microarrays or ddPCR. For 83 out of 87 target genes studied (95%), the agreement between the CoreExome microarrays and NanoString nCounter was characterized by PABAK values < 0.75, except for MAGI3, PDGFRA, NKX2-1 and KDR genes (>0.75). The MET, HMGA2, KDR, C8orf4, PAX9, CDK6, and CCND2 genes had the highest agreement among all three approaches. **Conclusions**: Therefore, to get a better idea of how to genotype an unknown CNV spectrum in tumor or normal tissue samples that are very different molecularly, it makes sense to use at least two CNV detection methods. One of them, like ddPCR, should be able to quantitatively confirm the results of the other.

## 1. Introduction

There are a group of genetic changes called gene copy number variations (CNVs) that show up at different rates in different parts of the genome of tumor cells [1,2]. Some estimates say that these changes, which mostly involve amplifications, insertions, deletions, and translocations, can rearrange DNA regions that are anywhere from 1 to 5 MB in size [3,4]. CNVs are elements of genomic instability and contribute to the survival of individual clones of neoplastic cells during tumor progression through disruption of transcriptional programs [5,6]. It is thought that CNVs found in tumors are more stable than expression biomarkers [7,8]. This means that studying the signatures of CNVs in different types of malignant neoplasms is a promising area for finding biomarkers that can tell us how a disease will progress. For instance, in a retrospective study [9], the genomic alterations correlated with the amount of circulating free DNA (cfDNA) and the frequency of occurrence of somatic allelic variants as well as the mutation tumor burden. The FDA approved the FoundationOne CDx Targeted Panel Clinical Assay (Foundation Medicine, Cambridge, MA, USA) for clinical use to look for problems in cfDNA taken from blood plasma and to find therapeutic CNV markers of how well the targeted drug therapy is working [10]. The Cancer Genome Atlas (TCGA) project looked at data from 33 different types of cancer and found typical tumor CNV signatures that were present in 97% of samples [2]. These signatures play a role in the development and growth of malignant neoplasms as well as chemotherapy resistance. This underlines the relevance of CNV detection to identify predictive or prognostic clinically relevant biomarkers. Another promising area of application of CNV data is the defining of molecular subtypes of solid tumors, which has particularly been reflected in several recent works [11,12,13]. However, it must be admitted that the results of similar studies are still far from being implemented in clinical practice.

Whole exome sequencing (WES) and whole genome sequencing (WGS) are NGS-based methods used to find gene number changes and larger chromosomal problems. In recent years, these kinds of analyses have been used a lot in preclinical studies to find CNV biomarkers linked to the development of many diseases [14,15,16]. Technological challenges for CNV detection in short-read sequencing and probe methods include extreme GC content and pseudogene homology, which make it difficult to unambiguously determine read alignment. The use of NGS panels is extremely convenient but requires high coverage in each position, special expensive analytical equipment, and adapted bioinformatics software for detecting CNVs in NGS ‘big data’. In the second case, there are a lot of different algorithms [17,18], which make it harder to use NGS analysis and determine what CNV data mean in everyday clinical practice.

There are several modern methods for CNV analysis, both for detecting aberrations in a single gene of interest and in a panel of genes. These methods can be conditionally divided into two types: a complete study of the genome or a significant part of it, and methods for the detection of clinically relevant CNVs in a number of targeted genome regions. High-throughput methods such as whole genome sequencing [19], microarrays [20], and comparative genomic hybridization (CGH) [21] allow for the detection of CNVs in tens to thousands of genes. Certain techniques, like NanoString CNV panels [22,23], PCR analysis in its different forms, and fluorescence in-situ hybridization (FISH) [24], can be used to specifically find CNVs.

FISH technology is a traditional method for detecting gene deletions and amplifications, which is laborious and time-consuming but still remains the gold standard for the analysis of clinically significant CNVs of gross chromosomal segments in many molecular genetic laboratories [25]. MLPA, in turn, is the gold standard for studying CNVs at gene size resolution [26].

The technology of microarrays is primarily designed to determine a large number of single nucleotide polymorphisms (SNPs). Illumina, for example, makes microarrays with different densities. The Infinium QC Array-24 has about 15,000 SNPs, while the Infinium Omni5-4 Kit has about 4.2 million SNPs [27]. In our study, CoreExome microarrays, containing about 550,000 SNPs and determining about 20 SNPs per one gene, were used [28]. However, several SNPs within a genome fragment determine its number of copies, potentially limiting the accuracy of such a determination. Thus, microarray technology makes it possible to detect CNVs within the exome with the expected moderate accuracy. Microarray technology is still the most common basic way to genotype a wide range of CNVs. However, not all platforms are equally good at finding different types of CNVs that are of interest [15]. Because of this, it is important to check the results of DNA microarray or NGS analysis using a different, highly repeatable method, like ddPCR [29]. This will lower the amount of false positive data when finding CNVs.

The NanoString nCounter v2 Cancer CN Assay (NanoString CNV) is a specially designed method for CNV detection in 87 genes associated with carcinogenesis [30]. It can be used to analyze a specific set of genes with high precision and, in particular, to validate CNV data obtained with hybridization technologies and microarrays [31].

Digital Droplet PCR (ddPCR) provides an accurate and absolute quantification of molecular targets with high sensitivity and reproducibility. The analytical sensitivity and specificity of the FISH and ddPCR are over 95% and 98%, respectively [32]. Recent studies have demonstrated the utility of different ddPCR versions for CNV detection [24,33].

The application of the above-mentioned methods has certain limitations. The results of high-throughput sequencing by synthesis, such as whole genome sequencing (WGS) and whole exome sequencing (WES), depend on the efficiency of the amplification of various genome regions and the length of reads, which limits the accuracy of regions with high sequence repeatability. The density distribution of polymorphisms in genome regions constrains the use of microarrays for SNP detection in CNV studies. Sequencing and analysis of microarrays make it possible to study a significant part of the genome, but they are highly dependent on bioinformatic methods that are used for CNV calling [34]. Techniques such as NanoString CNV assay and PCR analysis are expected to give higher accuracy of CNV detection in the regions studied, however their results depend on internal controls selected as regions with “normal” copy numbers [29]. In addition, these methods allow for the study of degraded DNA from FFPE blocks [30].

The methodical differences of the technologies on which CNV detection methods are based, along with the different degree of consistency between the results [35,36,37,38], actualize benchmarking studies on CNV detection, enabling this type of analysis to be conducted relatively quickly and with high accuracy.

According to the Global Cancer Observatory portal [39], ovarian cancer is the primary cause of death among gynecological cancers, with a five-year survival rate of 40–45% in the early stages (stages I–II) and 5–20% in the late stages (stages III–IV), respectively. High-grade ovarian serous carcinoma (HGSC) is the most aggressive subtype of ovarian cancer and is often diagnosed at advanced stages [40,41,42].

A bioinformatic analysis of TCGA datasets showed that ovarian cancer samples had the highest rates of CNV burden. There was also a lot of variation in the number of somatic copy number alterations (SCNA events) that happened in each sample [3]. Several studies [43,44] have demonstrated the identification of tumor-associated CNV biomarkers with prognostic or predictive value for ovarian cancer. Thus, the detection of CNVs in ovarian cancer samples is a variant of the analysis of a valuable way to discover specific genetic aberrations with potential clinical significance.

In this study, we want to compare how well three different methods find CNVs in paired samples of conditionally normal and ovarian cancer tumor tissues. We will also use statistics to see how consistent the results are. To detect CNVs, Infinium CoreExome microarrays (Illumina, Inc., San Diego, CA USA) and an nCounter v2 Cancer CN Assay panel (NanoString Technologies Inc., Seattle, WA, USA) were used. Digital droplet PCR QX200 (Bio-Rad Laboratories Inc., Hercules, CA, USA) was applied to verify the results obtained by these two methods.

## 2. Materials and Methods

### 2.1. Clinical Samples and Their Characterization

Thirteen patients were recruited to participate in a scientific study in the Federal State Budgetary Institution “National Medical Research Center for Obstetrics, Gynecology and Perinatology named after academician V.I. Kulakov” the Ministry of Healthcare of the Russian Federation. Of the 13 patients, 12 had a histologically confirmed diagnosis of high-grade serous carcinoma, and one had endometrioid carcinoma. For all the patients, sets of tumor samples and corresponding normal ovarian tissue were obtained. The samples were cryopreserved and then used for DNA extraction and molecular genetic studies. The characterization of clinical cases by disease stage, TNM classification, and immunohistochemical (IHC) results and parameters for assessing IHC markers are presented in Table A1 and Table A2 (Appendix A), respectively.

### 2.2. DNA Isolation

DNA for analysis was isolated from samples of fresh frozen tissues using the QIAamp DNA Mini Kit (Qiagen, Hilden, Germany). DNA quantification was performed using the Promega QuantiFluor ONE dsDNA kit (Promega Corporation, Madison, WI, USA) with a Quantus fluorimeter (Promega Corporation, Madison, WI, USA ). DNA purity was measured using NanoDrop 8000 (Thermo Fisher Scientific Inc., Waltham, MA, USA). For all the isolated DNA samples, the ratio of absorbance indices 260/280 was 1.8–2.0, the ratio 260/230 was 2.0–2.2.

### 2.3. CoreExome Microarray-Based Genotyping

Microarray-based genotyping was performed using the Infinium CoreExome-24 v1.3 BeadChip kit (Illumina, Inc., San Diego, CA, USA). The samples were processed using the Freedom EVO automated workstation (Tecan Group Ltd., Männedorf, Switzerland). Microarrays were scanned on the Illumina iScan System (Illumina, Inc., San Diego, CA, USA) with the AutoLoader 2.x autoloading module.

After scanning using the gtc2vcf plugin for bcftools V1.20 [45], vcf files were obtained from IDAT files. CNV was determined using the cnv plugin for bcftools [34]. The results were processed using R packages [46]. Data on the presence of CNV were also obtained using the cnvPartition CNV Analysis Plugin V3.2.0 and CNV Region Report Plug-in for Illumina GenomeStudio 2.0 in accordance with the manufacturer’s instructions.

### 2.4. Estimation of the Gene Copy Number Using a Panel nCounter v2 Cancer CN Assay (NanoString CNV)

The nCounter v2 Cancer CN Assay Kit for the nCounter FLEX Analysis System (NanoString Technologies Inc., Seattle, WA, USA) was used to estimate the number of gene copies. DNA was fragmented with the AluI enzyme and hybridized with the probes according to the manufacturer’s protocol MAN-10093-01 (NanoString Technologies Inc., Seattle, WA, USA). A total of 300 ng of DNA was used for hybridization; the incubation time was 18 h at 65 °C. The samples were then placed in the nCounter Prep Station for post-hybridization processing and immobilization on the cartridge.

The cartridge with the applied samples was transferred to the nCounter Digital Analyzer for scanning. A Prep Station and Digital Analyzer were used according to the user guide MAN-C0035 nCounter Analysis System MAX/FLEX. The nCounter data were processed using the nSolver 4.0 Analysis Software (NanoString Technologies Inc., Seattle, WA, USA) in accordance with the user manual MAN-C0019-08. Using the nSolver software, the signals from the samples were normalized, and data were obtained on the number of gene copies in the studied samples relative to the standard sample from the nCounter v2 Cancer CN kit.

### 2.5. PCR Analysis

The reference sequences of 15 genes of interest and 2 reference genes were found in the NCBI USA GenBank database [47] (https://www.ncbi.nlm.nih.gov/genbank/ (accessed on 30 August 2024)). The ALB and RPP30 genes, which are commonly used for the quantification of human genomic DNA [48], were selected as references. The primers and probes were designed according to the basic requirements for primers and TaqMan probes [49].

The selected primer and probe sequences were then checked in silico for specificity of annealing using the NCBI Primer BLAST online service (https://www.ncbi.nlm.nih.gov/tools/primer-blast/ (accessed on 30 August 2024)) [50]. The primer and probe sequences are shown in Table A3 (Appendix B). To prepare the TaqMan probes, FAM (fluorescein) or R6G (rhodamine 6G) was incorporated as a fluorophore at the 5′ end of a specific primer, and BHQ1 (Black Hole Quencher 1) was used as a fluorescence quencher at the 3′ end.

The melting and annealing temperatures of the primers and TaqMan probes were calculated using the online tool OligoAnalyzer IDT [51] (https://www.idtdna.com/pages/tools/oligoanalyzer (accessed on 30 August 2024)). This tool was also used to estimate the probability of the formation of secondary structures such as hairpins, homo- and heterodimers.

In-vitro functioning of the selected primer pairs and optimal amplification conditions were checked by symmetric gradient PCR on a CFX96 Touch real-time platform (Bio-Rad Laboratories Inc., Hercules, CA, USA). All the primer pairs were validated using melting curves. The size of the target fragments obtained by PCR analysis and the presence/absence of non-specific products were evaluated using the Agilent 4200 TapeStation system (Agilent Technologies, Inc., Santa Clara, CA, USA).

Digital droplet PCR was performed using the QX200 droplet digital PCR system (Bio-Rad Laboratories, Inc., USA). The ddPCR reaction mixture with probes consisted of 10 μL of 2x ddPCR Supermix for Probes (Bio-Rad Laboratories Inc., Hercules, CA, USA), 10 ng of tested DNA, 900 nM primers, and 250 nM hydrolysis probes for the target and reference genes in a final volume of 20 μL. The ddPCR reaction mixture with EvaGreen consisted of 10 μL of 2× ddPCR EvaGreen Supermix (Bio-Rad Laboratories Inc., Hercules, CA, USA), 10 ng of tested DNA, and 100 nM primers for target and reference genes in a final volume of 20 μL. The entire reaction mixture was loaded into the disposable plastic cartridge (Bio-Rad Laboratories, Inc., USA) along with 70 μL of droplet generation oil (Bio-Rad Laboratories Inc., Hercules, CA, USA) and placed into the droplet generator (Bio-Rad Laboratories Inc., Hercules, CA, USA). After processing, the droplets generated from each sample were transferred to a 96-well PCR plate. After PCR, the plate was loaded into the droplet reader (Bio-Rad Laboratories Inc., Hercules, CA, USA); the acquired data were analyzed using QuantaSoft software V1.0 (Bio-Rad Laboratories, Inc., USA). PCR amplification was carried out on a C1000 Touch thermal cycler (Bio-Rad Laboratories Inc., Hercules, CA USA) using the following conditions described in Table A4 and Table A5 (Appendix C). After the end of PCR, the plate was loaded into a droplet reader, and the data obtained were analyzed using the QuantaSoft software.

### 2.6. Biostatistics Data Analysis

According to the data yielded by the microarray platform, the variable of interest is divided into three categories: deletion, normal copy number state, and amplification, irrespective of the computational protocol (CNVpartition or bcftools). On the same note, NanoString CNV technology or PCR-based methods yield quantitative gene copy number values.

The conversion rule of copy number values from quantitative to integer for the NanoString CNV and PCR-analysis results was defined in keeping with the need to assess the consistency between quantitative and integer variables. It should be noted that often it is difficult to obtain the integer values of the gene copy number in the cancer tissue. This is caused by intratumor heterogeneity, which results in the existence of several cellular subgroups per tissue sample with differing copy number changes in each one [52].

Since the results were compared with a method that produces only integer values (microarrays), the ranges of this translation were determined for this task in accordance with the developer’s recommendations. Data Analysis Guidelines for Copy Number Variation (CNV) recommends considering all values in the range ±0.4 as the integer, while other values that exceed such a range should be studied on a separate basis or excluded from the analysis [53].

Since it was necessary to compare the results in pairs and each individual value of the copy number played a smaller role, the following rules of conversion were defined: copy number values between 1.5 and 2.5, inclusively, were considered as 2 (conditionally normal gene copy number), and values lower than 1.5 and greater than 2.5 were accepted as the deletion and amplification, respectively. These rules were used to convert the values obtained both by the NanoString CNV method and by PCR-based techniques.

To assess the reliability of consistency between integer or even qualitative variables obtained by two diagnostic methods, we calculated the prevalence-adjusted and bias-adjusted Kappa coefficient (PABAK) [54,55] by the epiR library written for the R programming language (epiR version 2.0.50) [56]. The PABAK value close to zero indicates a chance of consistency, while the PABAK value close to −1 means that nearly all the results of the first diagnostic method are in direct contradiction with the other one and vice versa. The PABAK value close to 1 indicates perfect consistency.

If the results of two diagnostic methods were quantitative variables, then Passing–Bablok regression was used to analyze the consistency. The reliability of consistency between quantitative variables had been assessed by; regression using the mcr library written for the R programming language [57]. This method is one of the most robust linear regression algorithms [58] and can be described as the equation:$$y_{i}=k*x_{i}+b$$
where $x_{i}$ and $y_{i}$ are the gene copy number values for i-th observation that were obtained by the first and second method, respectively. The slope coefficient k corresponds to a proportional bias and the intercept b corresponds to a constant bias. The presence of value one inside the 95% confidence interval for k means no proportional difference between two methods and the presence of value zero inside the 95% confidence interval for b means no constant difference between two methods.

The discrepancy between values obtained by both methods could be unacceptably large even without significant proportional and constant biases [59]. It should be noted that consistency and interchangeability between two methods is influenced by the scatter of points around the line of complete consistency, i.e., y=x. Therefore, in addition to the Passing–Bablok regression, the following measure was calculated:$$d_{i}=\frac{\left|y_{i}-x_{i}\right|}{\max\left(y_{i},\;\;x_{i}\right)}$$
where $x_{i}$ and $y_{i}$ are the gene copy number values that were obtained by both methods.

To evaluate the statistical significance of the difference between the copy number profiles of normal and tumor tissue, we used the Wilcoxon test for the quantitatively paired samples. The same pair of samples had been studied using two or more methods of copy number analysis, so for such a pair, the respective number of statistical testing procedures should be performed. The harmonic mean of the resulting set of *p*-values was calculated. To counteract the multiple testing problem, all the obtained harmonic means of *p*-values were adjusted according to the Holm method.

The study utilized R programming language (version 4.2.1) and RStudio IDE (version 2022.02.3.492).

## 3. Results

### 3.1. Comparison of Results of Gene Copy-Number Assessment

In the first stage of the study, the gene copy number was determined using the NanoString CNV panel and microarrays. The CNVs on microarrays were determined by the same coordinates, which are targeted by labels in the NanoString CNV panel. Figure A2 (Appendix G) distinguishes at least three groups of genes: MAGI3, PDGFRA, NKX2-1, MET, and KDR are genes with positive consistency (PABAK values closer to one); CRKL, BBC3, BCL2L1, AR, and CDK4 are genes with medium consistency (PABAK values close to zero); and ERBB2, BRCA1, MAPK7, FGFR1, and ITGB4 are genes with negative consistency (PABAK values less than 0).

After analyzing the consistency of the CNV detection results using NanoString CNV panels and microarrays in the second stage of the study, we selected gene panel no. 1 (Table A6 (Appendix D)) for the ddPCR verification of the identified CNVs. The specifics of the primer design and ddPCR settings are described in the Materials and Methods section, as well as in Appendix C (Table A4 and Table A5). The research plan is presented in Table A6 (Appendix D), and the results of the experiments are shown in Figure 1 (see also Figure A2 (Appendix G)).

### 3.2. CNV Detection by Digital Droplet PCR

The next step was to assess the copy number of the target genes relative to the reference genes RPP30 and ALB using Bio-Rad QX200 digital droplet PCR. The ddPCR results were used to evaluate the performance of the primer sets. To accomplish this, the gene copy number values in normal tissue were evaluated. The median value among patients falling within the range of 1.5 to 2.5 was taken as the normal value of the copy number (Figure 2). We detected gene copy number variations using four different versions of PCR analysis: two reference genes and two different PCR protocols. If the median copy number values of a single gene in normal tissue samples were less than 1.5 and more than 2.5, then such a group of observations was considered abnormal and excluded from further consideration (Figure 2). Additionally, we decided not to consider the results of the gene copy number assessment, which used albumin (ALB) as a reference gene, because a significant portion of the gene copy number values in normal tissue samples fall outside the range of 1.5 to 2.5.

### 3.3. The Consistency between Three Methods of CNV Detection

Since the results of the microarray copy number estimation are in an integer format, we began by comparing the NanoString CNV panel and PCR results, which enabled us to quantify the copy number of the genes. Table 1 presents the consistency analysis of CNV detection using the NanoString CNV panel and ddPCR analysis. Only the CCND2 gene showed a statistically significant change in copy number variation. The NanoString CNV panel and PCR analysis detected an increase in the copy number values of this gene in tumor tissues compared to normal tissues. Despite the small sample size, when comparing the results of PCR analysis (tumor/normal), statistically significant differences in the gene copy number were obtained; when comparing the results of the NanoString CNV panel (tumor/normal), statistical differences were noted at the trend level of statistical significance (0.05 < *p* < 0.1). As for the other genes in Table 1, there were no statistically significant changes in the gene copy number variations.

Figure 3 presents three pairwise comparisons of the CNV detection results. The dashed sloping line in each of the graphs indicates the ideal situation in which the application of both methods on the same set of samples leads to identical results. The solid sloping line represents the resulting regression model. In the table below, there are data indicating the absence (x) or the presence (o) of a statistically significant proportional (k) and systematic errors (b). The results of calculating the magnitude of the relative discrepancy *d_i_* are displayed. We determined the value of *d_i_* for all samples considered in the study, allowing us to judge the consistency of the two methods based on the characteristics of the formed sample. The closer the median, first, and third quartiles of the sample d to zero, the better the consistency of the two methods.

As you can see in Figure 3, the two types of PCR analysis—the one using the EvaGreen intercalating dye and the one using TaqMan probes (Probes)—are the most consistent and have the smallest amount of difference between them. The NanoString CNV method is characterized by low consistency with methods for detecting copy number variations using PCR analysis. A pairwise comparison of the NanoString CNV panel with the PCR analysis reveals a systematic error. The gene copy numbers obtained by the PCR analysis were generally lower than those obtained by the NanoString CNV panel.

For further pairwise comparisons between three lab approaches and their variations, quantitative values were converted to categorical values according to the scheme described in the Materials and Methods section. A graphical panoramic view of consistency between the results of the CNV detection by three different methods is shown in Figure 1, and more detailed data are presented in Figure A2 (Appendix G). As shown in Figure 1, the results obtained by the two versions of CoreExome microarrays are the most consistent with the bcftools and CNVpartition calling protocols, respectively. In addition, it is noticeable that the results of the PCR analysis are in good consistency with the results of the CoreExome microarrays using the CNVpartition calling protocol. On the contrary, the results obtained by the NanoString CNV panel are satisfactory and consistent with those obtained by all other methods. For pairwise comparisons, the MET, HMGA2, KDR, PAX9, CDK6, and CCND2 genes had the best consistency values. This was true no matter what method was used to find CNVs.

### 3.4. Results of CNV Detection in Ovarian Cancer Samples Based on ddPCR Data

It follows from Figure 4 that the total number of detected deletions is ≈1.6 (25/16) times higher than the number of amplifications of nine target genes in 12 HGSC samples. The deletions were mostly found in the CDKN2A (cyclin-dependent kinase inhibitor 2A) and PTEN (phosphatase and tensin homolog) genes, with 7 and 8 events per 12 samples, respectively. Copy number amplifications were found mainly in CCND2 (cyclin D2) and PTPRD (protein tyrosine phosphatase receptor type D) genes—3 and 4 events per 12 samples for each gene, respectively (Figure 4).

In tumor samples, the CNVs of other genes were irregularly distributed. For the TCGA serous cystadenocarcinoma ovarian dataset (n=617), cBioPortal data show that 5.9% of CDKN2A genes and 5.7% of PTEN genes have copy number variations. Given that the mutant TP53 gene (which codes for the tumor suppressor p53) is found in about 87% of all ovarian cancer samples, it is likely that the CDKN2A and PTEN genes are also changed. For instance, we found that deep deletions of the CDKN2A gene correlate with TP53 mutational signatures and worse survival in breast cancer subjects [60]. Thus, we attempted to search for the potential associations between the most frequent CNVs of CDKN2A, PTEN, CCND2, and PTPRD genes with clinical data (Table A1 and Table A2, Appendix A). For this, a relatively ‘homogeneous’ subset of nine HGSC samples (SN30, SN32, SN36, SN38, SN44, SN46, SN50, SN54, SN56), having a similar tumor stage, TNM classification (T3N0(1)M0(1)), as well as Ki-67 and p53 IHC-staining patterns (Table A1 and Table A2, Appendix A), were analyzed. Notably, deletions of the CDKN2A and PTEN genes were observed in 3 patients (SN38, SN50, and SN54) with late stage cancer (IVb) and metastasis (T3NxM1), which may warrant further investigation. Apparently, the small sample size did not allow us to find more associations between the number of detected CNVs of target genes and other clinical data; for example, the number of CNVs in stage IIIc ovarian cancer cases ranged from 0 to 4. This was similar to the number of CNVs seen in samples from stages Ia and IIa (Figure 4), which were 2 and 3. Further, there was no correlation (Spearman’s rank correlation coefficient = 0.16, *p*-value > 0.1) between the total number of CNVs of all nine genes under consideration (Figure 4) and the ages of 12 subjects with HGSC.

## 4. Discussion

In general, works on the benchmarking of methods are interesting since it is known that, depending on the source of the data acquisition, their consistency can vary significantly. This was revealed by a pairwise comparison of information on gene copy loss extracted from the Hapmap Project and 1000 Genome Project repositories (less than 30% overlap) [61]. In our study, the NanoString CNV panel was used to genotype CNVs in tumor samples obtained from patients with high-grade serous carcinoma. This commercial product has recently appeared on the biotechnological market and is suitable for targeted multiplex analysis of gene abnormalities associated with oncogenic molecular pathways, so the researchers’ interest is directed at clarifying the limits of its application. Recently, it was shown that comparing the results of the NanoString CNV panel (20-gene signature for copy number value detection) with the best ways to measure gene expression, like IHC analysis and in-situ hybridization (FISH), showed a good level of agreement (κ-value = 0.35) in the HER2 copy number variation from biopsy gastric cancer tissues (FFPE blocks) with the results of IHC/FISH, which had sensitivity and specificity values of 66.7% and 85.2%, respectively [62]. Data on benchmarking the results of the study using the NanoString CNV panel, DNA-microarray technologies, or ddPCR to study the CNV profile of HGSC ovarian cancer is not yet available, highlighting the novelty of the obtained results.

In the study using CoreExome microarrays, the same raw data were processed using bcftools and CNVpartition tools. The median PABAK value was 0.75 (Figure 5), which gives us an idea of what the theoretical limit of consistency might be for these comparisons. This is because we should obtain more consistent data when we compare different ways of processing the same raw data than when we compare different biological methods.

The results of the copy number assessment using CNVpartition show better consistency with each of the other methods than the values obtained using bcftools. The gene copy number detection using the NanoString CNV panel shows good consistency rates, except for the comparison with bcftools and ddPCR analysis using probes. The results most comparable to all other methods were shown by the CoreExome-CNV partition and ddPCR with intercalating dye. Most comparisons fall within the PABAK value range of 0.61–0.80, which is considered a substantial agreement. In our opinion, values below 0.61 indicate an insufficient level of result consistency. Ideally, the consistency level should exceed PABAK 0.8. In our opinion, this result is extremely difficult to achieve using off-the-shelf panels, but this level of consistency appears realistic when developing customized labels for targeted approaches such as ddPCR.

Copy number assessment can be inconsistent for a number of reasons, such as the genetic structure of the area being studied, the design of the ddPCR probe, and the way the reaction is set up. Another factor we take into account is the small number of comparisons, i.e., ten genes. The consistency scores between samples analyzed with the NanoString CNV panel and various CoreExome microarray processing tools show a significant difference: the median value of PABAK differs by two-fold (0.31) when compared with bcftools and by 0.62 when compared with the CNV partition tool. Relatively low consistency between the two ddPCR methods can be expected due to the small number of compared genes, and as a result, each individual low consistency value contributes more to the median value. It should be noted that the study was carried out on real tumor and normal tissue samples, meaning that there was no known unambiguous copy number value for each gene, in contrast to studies on well-studied cell lines.

Researchers can use the results of gene copy number variations in ovarian cancer tissue samples not only to characterize the genetic profile of tumors, but also to determine the potential prognostic significance of the identified CNVs. In particular, previous studies found that risk assessment based on CNV detection was more valid in predicting overall survival in ovarian serous adenocarcinoma. Thus, the ten-year CNV risk score had an AUC value of 0.747 [43].

Optimization of the gene panel for performing CNV genotyping and data interpretation is critical due to the high initial heterogeneity of tumors from different subjects in the cohort. For instance, a study [63] selected a panel of 34 genes based on tumor heterogeneity to catalog CNVs in a cohort of 96 patients with HGSC. The nCounter v2 Cancer CN Assay panel consists of 87 target genes and is intended for CNV genotyping of tumors with various profiles of aberrations in the PIK3CA, AKT, PTEN, BRCA, ERBB2, and MYC genes. The NanoString CNV panel can be used to test people with different types of cancer because it covers 55% of the genes where mutations are most common, according to data from the Cancer Gene Census [64].

The frequencies of occurrence (>2%) of amplifications and deletions of the genes of the NanoString CNV panel were observed in a cohort of patients with ovary serous carcinoma (568 samples) [65], where 23 and 4 genes, respectively, accounted for 31% of all genes in the panel (27 of 87 genes). Thus, it can be concluded that this NanoString CNV panel is suitable for detecting CNVs in ovarian cancer samples. Our study’s statistical analysis revealed that seven target genes (MET, HMGA2, KDR, C8orf4 (TCIM), PAX9, CDK6, CCND2) (a gene panel No. 2) exhibited the highest consistency between the results of CNV detection by different methods (DNA microarray, NanoString CNV, and ddPCR analysis). The biological features of these genes as well as their potential prognostic value in ovarian cancer are discussed in Appendix F, which includes a text subsection, Table A8, Figure A1, and the following literature references [43,65,66,67,68,69,70,71,72,73,74,75,76,77,78,79,80,81,82,83,84,85,86,87].

Our benchmarking study focused on assessing amplifications and deletions as two major categories of CNVs. At the same time, the repertoire of CNVs is represented by a wider spectrum of subcategories due to the high inter- and/or intra-patient tumor heterogeneity and complex nature of CNV combinations [2]. In the current study, we did not differentiate such subcategories by the experimental and bioinformatics methods used, but some of them are mentioned below. The prevalence of CNVs as indicators of genome instability was investigated by Marco and colleagues [88], who identified 201 chromosomal regions and 3300 genes subjected to copy number (CN) alterations in patients with serous ovarian cancer. Out of 576 genes, CN gains comprised 530 low-level and 46 high-level CN gains, while CN loss affected 2724 genes (2454 heterozygous and 270 homozygous deletions, respectively), indicating a 5-fold higher frequency of CN losses compared to CN gains. Cheng and colleagues examined the distribution of homozygous deletions in different cancer types [89]. It was shown for ovarian cancer that deletions ranged from 0 to 6 per case and had a median fragment length of 207 kb per case. Deletion peaks occurred primarily in the tumor suppressor gene CDKN2A, followed by RB1 (RB transcriptional corepressor 1) and then the PTEN gene. This fits with the fact that about 70% of the CDKN2A and PTEN genes found in HGSC samples were deleted, which is higher than the rate of deletion seen in other genes in our study.

In type II high-grade serous ovarian cancer, there is genetic instability and a lot of DNA CN gains or losses. Almost all tumors also have TP53 mutations [90]. Typically, loss of heterozygosity (LoH) of the wild-type allele of tumor suppressor genes (TSGs) can also be caused by CN losses. On average, across all cancers, 16% of genes underwent LoH, with a median of about 30% in ovarian cancer. Rates of LoH were no lower for cell-essential genes, which also included some TSGs, relative to the rest of the genome; tumors harbored an average of 189 essential genes with LoH [91]. Given that the frequency of LoH in tumors is one order of magnitude higher than that of homozygous deletions [88], we can use a specific pattern of LoH to predict disease prognosis. For instance, two relatively small regions of chr 19 (8.0–8.8 and 51.5–53.0 Mbp regions, which include 19 and 37 genes, respectively) exhibited the most significant differences in LoH. Functional loss of the TP53 gene due to mutations in chr 17 found in 32% patients and LoH in chr 19 were associated with serous ovarian cancer recurrence [92]. Also, it is known that turning off both alleles (biallelic inactivation) of TSGs makes ovarian cancer grow and spread a lot [93,94]. Ryland and colleagues [95] suggested several models of non-random accumulation of LoH in ovarian cancer genomes. Among these models are: the classic two-hit hypothesis, which involves high-frequency biallelic genetic inactivation of TSG; the epigenetic two-hit hypothesis, which involves high-frequency biallelic inactivation through methylation and LoH; the multiple alternate gene biallelic inactivation, which involves low-frequency gene disruption; the hap-lo-insufficiency model, which involves single copy gene disruption; and the modified two-hit hypothesis, which involves a reduction to homozygosity of predisposition alleles. These mean that there are more than one, possibly overlapping, ways in which different types of CN changes can lead to the development of tumorigenic traits. In addition, it is worth noting that the novel version of inter-laboratory classification concordance using the American College of Medical Genetics and Genomics (ACMG) and Clinical Genome Resources (ClinGen) CNV scoring metrics system was adapted for potential clinical use, with an overall complete concordance of up to 85% across laboratories [96].

## 5. Conclusions

The analysis of statistical consistency between the results of three methods of CNV genotyping allows us to make the following conclusions: (a) the use of one method cannot be considered sufficient to unambiguously detect CNVs; (b) data obtained by panoramic detection of CNVs using microarrays or NanoString CNV panels (or other methods of broad genome analysis for the presence of structural alterations) should be verified with a point quantitative analysis of selected CNVs, for example, with digital droplet PCR; and (c) most pairwise comparisons gave a PABAK value of >0.6, which is considered acceptable [55] and indicates good consistency between the results of the methods used; however, for a more detailed study of the CNVs of a single gene, it is reasonable to rely on consensus results obtained by at least two methods.

## Figures and Tables

**Figure 1 cancers-16-03252-f001:**
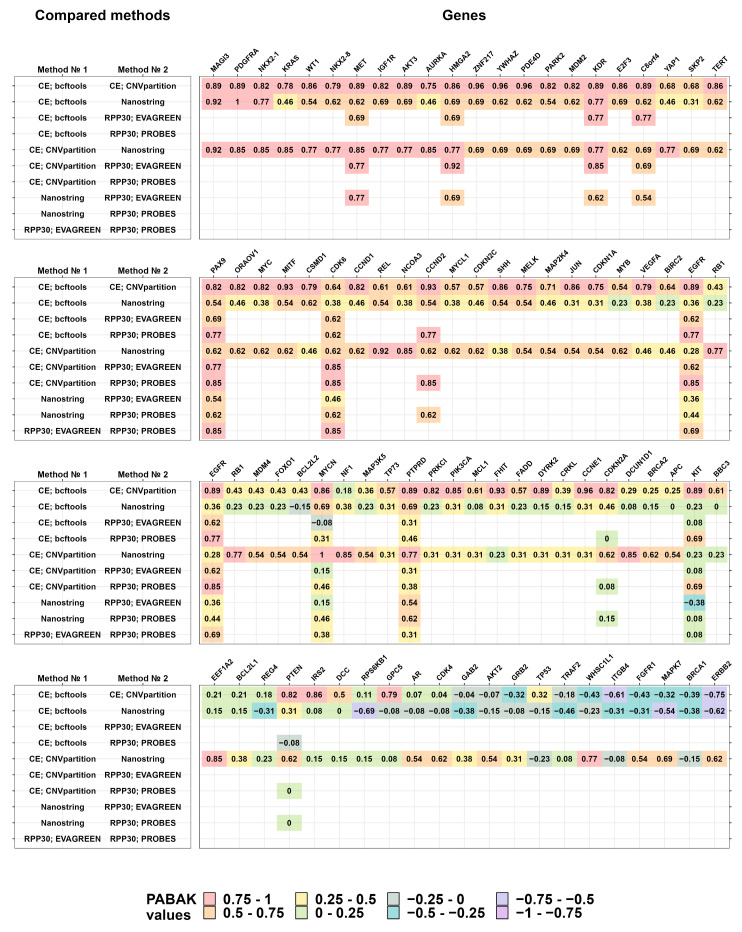
The consistency between the methods of copy number identification measured by prevalence-adjusted and bias-adjusted Kappa coefficients (PABAK). The three wide panels are the tables whose rows correspond to the compared methods and whose columns correspond to the studied genes. The sort order is decreasing according to the median PABAK value per gene, from highest to lowest.

**Figure 2 cancers-16-03252-f002:**
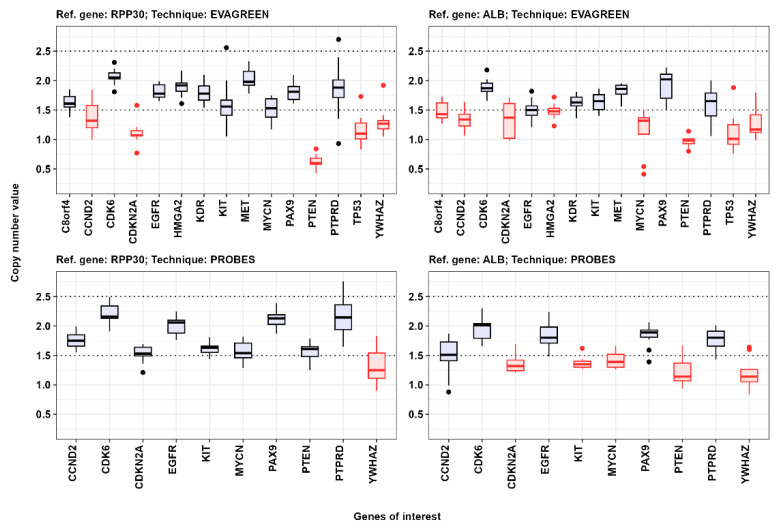
Preliminary descriptive statistics of PCR-analysis results for normal tissue. Copy number values are presented for the genes of interest. The red color depicts groups of observations that were excluded from further consideration.

**Figure 3 cancers-16-03252-f003:**
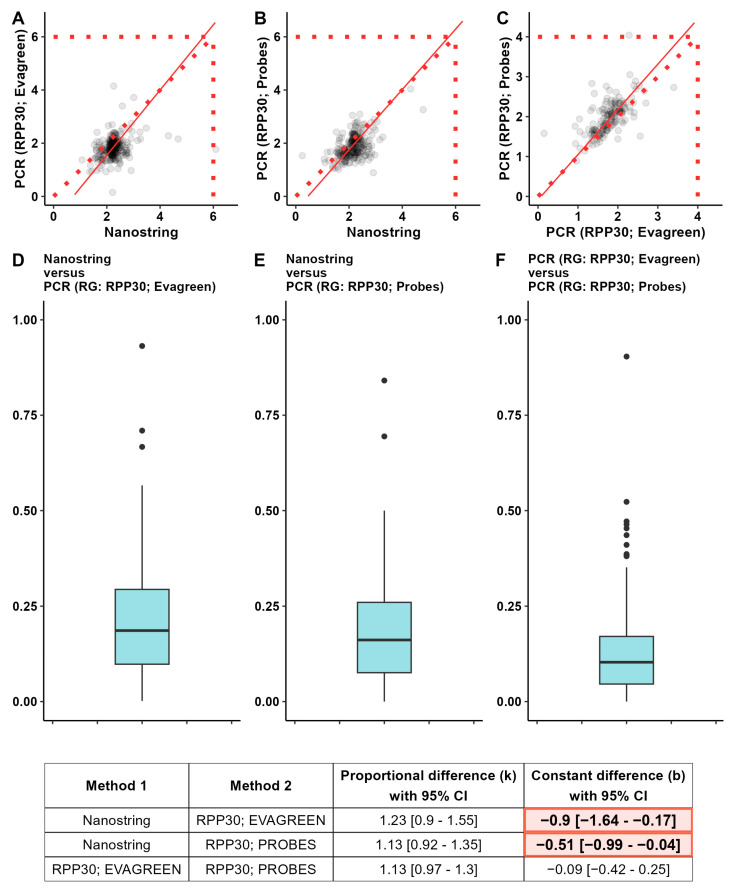
The consistency between the methods of copy number identification measured by Passing–Bablock regression and by the magnitude of the relative discrepancy *d_i_*. Three methods were considered: the NanoString CNV panel, the PCR-based technique with TaqMan probes, and the PCR-based technique with EvaGreen dye. RPP30 has been used for both PCR analyses as a reference gene. Panels (**A**–**C**) are the scatterplots depicting the results of the Passing–Bablock regression. The panels (**D**–**F**) are the boxplots representing the distribution of relative discrepancy *d_i_*. The lower the *d_i_* is to zero, the better the consistency of the two methods. The coefficients of the Passing–Bablock regression models are presented in the bottom table, along with their respective 95% confidence intervals. The shaded cells represent the observed constant differences between the compared results of the two methods. The abbreviation RG means «reference gene».

**Figure 4 cancers-16-03252-f004:**
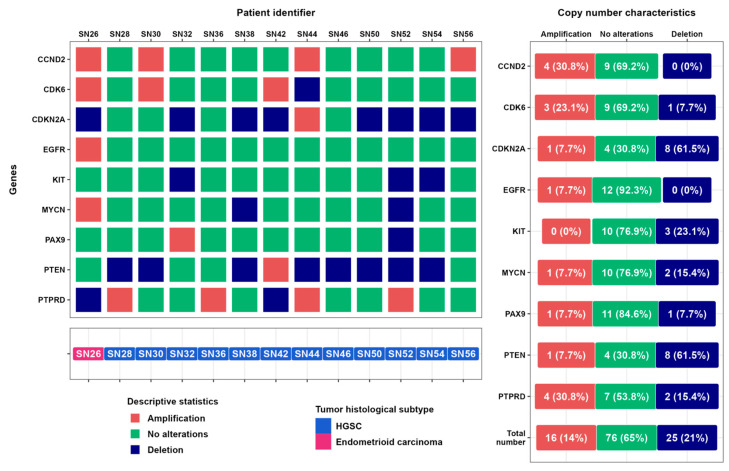
Changes in copy number values of genes of pane1 according to the PCR analysis. We used a PCR-based technique with TaqMan probes and RPP30 as a reference gene. The horizontal axis corresponds to the patient identifier, and the vertical axis to the genes of interest.

**Figure 5 cancers-16-03252-f005:**
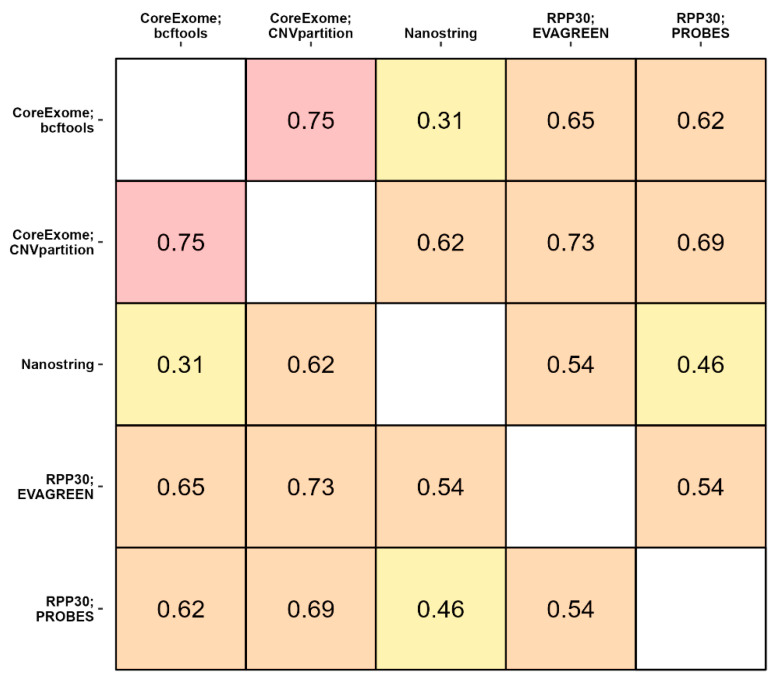
Pairwise consistency between the methods of copy number identification. The prevalence-adjusted and bias-adjusted Kappa coefficient (PABAK) has been used as a measure of consistency.

**Table 1 cancers-16-03252-t001:** Data on copy number variations in tumor and normal tissues *.

Genes	*p*-Value PCR EvaGreen	*p*-Value PCR Probes	*p*-Value NanoString CNV	*p*-Value	*p*-Value (Corrected)
*C8orf4*	0.972	No data	0.807	0.882	1
*CCND2*	No data	0.003	0.092	0.006	0.042
*CDK6*	0.497	0.216	1	0.393	1
*HMGA2*	0.505	No data	0.168	0.252	1
*KDR*	0.685	No data	0.363	0.475	1
*MET*	0.893	No data	0.421	0.573	1
*PAX9*	0.685	0.542	0.168	0.324	1

* the full description of the statistics is shown in Table A7 (Appendix E).

## Data Availability

The NanoString CNV panel raw data results are available under NCBI GEO accession GSE244329. Our code is available at Figshare.com depository: (https://figshare.com/s/b4c44d9f9e3ce7e2a74d (accessed on 20 September 2024)).

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
