# Peer review of "Benchmarking of Approaches for Gene Copy-Number Variation Analysis and Its Utility for Genetic Aberration Detection in High-Grade Serous Ovarian Carcinomas"

_cancers, 2024, doi:10.3390/cancers16193252_

Round 1

Reviewer 1 Report

Comments and Suggestions for Authors

Verification of CNV calling with an orthogonal method has advantage in precise clinical interpretations, but it increases the costs significantly. This urges bechmarking of the available techniques in real clinical applications, testing on tumor samples with high CNV characteristics. The authors carried out a series of carefully designed original research work to detect the pairwise consistency between three CNV detection methods in ovarian cancer samples. They tested commercial SNP-array and capture-array based methods, as well as ddPCR. They showed off, that not only the biological genotyping method, but also the data analysis are crucially important with respect of correct CNV interpretation. They revealed, that CoreExome microarrays with CNVpartition analysis showed the best consistency with each of the other methods. Harnessing the CNV data obtained from ddPCR they performed genotype-phenotype statistical correlations between CNV patterns and clinical stage of tumors. 

Minor corrections:

Please, check the whole text for formulation mistakes: there are some sentences, that are accidentally repeated or left uncorrected (Lines 59-60, 65-67).

Throughout the text: I suggest to use ‘versions’ instead of ‘variants’ of analysis methods.

Lines 73-77: Emphasize pseudogene interference and variable GC contents as an inflating role of CNV-detection, especially at capture-based enrichments.

The demand for high coverage in each position, rather than analytical tools make CNV analysis expensive in NGS methods.

Line 85 and 87: The resolution of FISH is amenable for rearrangements of gross chromosomal segments (megabase length) rather than genes. The traditional method for detecting gene deletions and amplifications is MLPA. Please, mention also MLPA technique, as a gold standard method for CNV detection.

Line 134: burden (without hyphen)

Line 137: Thus, the detection of CNVs in ovarian cancer samples is a variant of the analysis of a valuable way to discover specific genetic aberrations with potential clinical significance.

Line 191: How did you select for the 15 genes you analyzed by PCR methods? Are the CNV of these more relevant in ovarian cancer?

Line 175: New subheading

In Figure 2: The whiskers plots arise from how many data points? 13 normal samples?

Line 251: decimals with dot instead of comma

Line 297: with medium consistency (?)

Section 3.2: Where do the normal pairs of the ovarian tumor samples come from? It is likely, that there is CNV already in the normal tissue in the neighborhood of the tumor. If you wanted to control the results with surely 2-copy reference, would not it be better to compare to germline DNA?

Line 386: based on ddPCR data

General questions:

Both mentioned genotyping platforms (SNP-array and probe capture-based) are carefully designed for high-throughput CNV genotyping. Are these individually not sufficient for correct CNV pattern detections? In other words: does the application of an orthogonal method provide indispensable additional information with respect to clinical and pathological subclassification of tumor samples as a function of CNV patterns? Precise CNV calling in monogenic disorders is of great importance. In the case of CNV patterns, does it have notable benefit to decipher the CNV in single gene certainty?

If the analytical sensitivity and specificity of ddPCR are over 95% and 98%, is not it sufficient as a standalone method for the discovery of characteristic patterns?

It is notable and interesting, that there is modest consistency (0.54) between the two versions (probe-based and EvaGreen-based) of ddPCR. These are practically the same biological methods, show difference only in amplification detection technique.

In section 3.3 you wrote, that only CCND2 locus showed significant increase, when compared ddPCR and NanoString technology, but in 3.4 you showed off CNV gain and loss in a series of other genes with ddPCR method alone. Does it mean, that NanoString was not able to detect the majority of the highlighted CNVs and less amenable for clinical interpretations?

Author Response

Thank you for your valuable feedback! Below are the answers to each question:

Comment 1: Please, check the whole text for formulation mistakes: there are some sentences, that are accidentally repeated or left uncorrected (Lines 59-60, 65-67).

Response 1: We checked the text again and made some adjustments.

Comment 2: Throughout the text: I suggest to use ‘versions’ instead of ‘variants’ of analysis methods.

Response 2: Changed the wording to "versions".

Comment 3: Lines 73-77: Emphasize pseudogene interference and variable GC contents as an inflating role of CNV-detection, especially at capture-based enrichments.

The demand for high coverage in each position, rather than analytical tools make CNV analysis expensive in NGS methods.

Response 3: Added information about the influence of GC composition and pseudogenes, as well as the need for high coverage.

Comment 4: Line 85 and 87: The resolution of FISH is amenable for rearrangements of gross chromosomal segments (megabase length) rather than genes. The traditional method for detecting gene deletions and amplifications is MLPA. Please, mention also MLPA technique, as a gold standard method for CNV detection.

Response 4: FISH resolution has been refined and MLPA has been added.

Comment 5: Line 134: burden (without hyphen)

Response 5: Corrected.

Comment 6: Line 137: Thus, the detection of CNVs in ovarian cancer samples is a variant of the analysis of a valuable way to discover specific genetic aberrations with potential clinical significance.

Response 6: Corrected.

Comment 7: Line 191: How did you select for the 15 genes you analyzed by PCR methods? Are the CNV of these more relevant in ovarian cancer?

Response 7: Genes for PCR were selected from 3 groups that showed positive, medium and negative consistency between the other two methods (nanostring and microarrays). The role of these genes in the development of tumor diseases was not used for selection.

Comment 8: Line 175: New subheading

Response 8: Corrected.

Comment 9: In Figure 2: The whiskers plots arise from how many data points? 13 normal samples?

Response 9: True, the data are based on 13 samples in all cases except for the CDK6 (ALB; EVAGREEN) gene, which had 12 observations.

Comment 10: Line 251: decimals with dot instead of comma

Response 10: Corrected.

Comment 11: Line 297: with medium consistency (?)

Response 11: Corrected.

Comment 12: Section 3.2: Where do the normal pairs of the ovarian tumor samples come from? It is likely, that there is CNV already in the normal tissue in the neighborhood of the tumor. If you wanted to control the results with surely 2-copy reference, would not it be better to compare to germline DNA?

Response 12: We absolutely agree that it would be ideal to use germline DNA. Unfortunately, we could not expand the list of materials received for the study. Therefore, we used the second-best option - DNA from the surrounding normal tissue, which is labeled and confirmed by a pathologist.

Comment 13: Line 386: based on ddPCR data

Response 13: Corrected.

General questions:

Comment 14: Both mentioned genotyping platforms (SNP-array and probe capture-based) are carefully designed for high-throughput CNV genotyping. Are these individually not sufficient for correct CNV pattern detections? In other words: does the application of an orthogonal method provide indispensable additional information with respect to clinical and pathological subclassification of tumor samples as a function of CNV patterns? Precise CNV calling in monogenic disorders is of great importance. In the case of CNV patterns, does it have notable benefit to decipher the CNV in single gene certainty?

Response 14:

In our opinion, SNP-array and probe capture-based do allow us to assess the overall picture of copy number changes in a sample and detect the presence of a trend. However, changes in the copy number of individual genes may play a more significant role in diagnostics, so we consider it important to more accurately determine CNV in some genes, for example, using ddPCR.

Comment 15: If the analytical sensitivity and specificity of ddPCR are over 95% and 98%, is not it sufficient as a standalone method for the discovery of characteristic patterns?

Response 15:

The ddPCR method itself allows for highly accurate detection of CNV, and generally allows for the development of a diagnostic panel. However, the study of several dozen genes in a large number of samples using ddPCR will be quite time-consuming and labor-intensive, so there are more productive methods for ongoing research.

Comment 16: It is notable and interesting, that there is modest consistency (0.54) between the two versions (probe-based and EvaGreen-based) of ddPCR. These are practically the same biological methods, show difference only in amplification detection technique.

Response 16:

The development of a CNV detection system using ddPCR requires adjustments to many factors, especially the primer sequence, to obtain reliable and repeatable results. Our objectives did not include detailed development of an assay panel, which may explain the discrepancy in the results in ddPCR.

Comment 17: In section 3.3 you wrote, that only CCND2 locus showed significant increase, when compared ddPCR and NanoString technology, but in 3.4 you showed off CNV gain and loss in a series of other genes with ddPCR method alone. Does it mean, that NanoString was not able to detect the majority of the highlighted CNVs and less amenable for clinical interpretations?

Response 17:

In Section 3.3 we noted that only the CCND2 gene has statistically significant CNV when tested using ddPCR and NanoString. Both methods detect CNVs to a similar degree, but not every CNV detection is statistically significant.

Reviewer 2 Report

Comments and Suggestions for Authors

This is a research paper that attempts to shed more light into methods for the analysis of CNV in the genome of ovarian cancer patients and the differences between them that may allow for the more accurate use of techniques to analyse vast amounts of data.

This is a paper that needs only some minor revisions that are listed below:

1) Perform some minor editing in English mainly on a few occasions to correct some incomplete phrases.

2) Are there any other papers that perform similar analyses in other types of cancer examining the exact same set of tools as it has been done in this study?

3) Do the authors believe that the use of a larger number of samples (e.g. over 100) would confer a different type of result or disqualify any of the tools used in this study? Are there any examples like that on any type of cancer including ovarian cancer?

Comments on the Quality of English Language

Overall the English is of very good quality. There are only few correction requirements especially deleting any text repeats

Author Response

Thank you for your valuable feedback! Below are the answers to each question:

Comment 1: 1) Perform some minor editing in English mainly on a few occasions to correct some incomplete phrases.

Response 1: We checked the text again and made some adjustments.

Comment 2: 2) Are there any other papers that perform similar analyses in other types of cancer examining the exact same set of tools as it has been done in this study?

Response 2: At the time of preparation of the work, we could not find publications with a similar set of methods.

Comment 3: 3) Do the authors believe that the use of a larger number of samples (e.g. over 100) would confer a different type of result or disqualify any of the tools used in this study? Are there any examples like that on any type of cancer including ovarian cancer?

Response 3:

In the statistical analysis, we considered the copy number of a single gene in the tumor tissue of a single patient as one observation. Similarly, the copy number of a gene in normal tissue for a single patient was considered as one observation. Thus, the number of comparisons was quite large. We believe that increasing the sample size will clarify the picture, but will not change it radically. Oncology studies, according to our observations, either do not have very large sample sizes, or in the case of a larger sample, focus on a smaller number of laboratory studies.